# Identification of Similar Chinese Congou Black Teas Using an Electronic Tongue Combined with Pattern Recognition

**DOI:** 10.3390/molecules24244549

**Published:** 2019-12-12

**Authors:** Danyi Huang, Zhuang Bian, Qinli Qiu, Yinmao Wang, Dongmei Fan, Xiaochang Wang

**Affiliations:** Tea Research Institute, Zhejiang University, # 866 Yuhangtang Road, Hangzhou 310058, China; demiwaiting@zju.edu.cn (D.H.); 21816063@zju.edu.cn (Q.Q.);

**Keywords:** congou black tea, electronic tongue, type, grade, pattern recognition

## Abstract

It is very difficult for humans to distinguish between two kinds of black tea obtained with similar processing technology. In this paper, an electronic tongue was used to discriminate samples of seven different grades of two types of Chinese Congou black tea. The type of black tea was identified by principal component analysis and discriminant analysis. The latter showed better results. The samples of the two types of black tea distributed on the two sides of the region graph were obtained from discriminant analysis, according to tea type. For grade discrimination, we determined grade prediction models for each tea type by partial least-squares analysis; the coefficients of determination of the prediction models were both above 0.95. Discriminant analysis separated each sample in region graph depending on its grade and displayed a classification accuracy of 98.20% by cross-validation. The back-propagation neural network showed that the grade prediction accuracy for all samples was 95.00%. Discriminant analysis could successfully distinguish tea types and grades. As a complement, the models of the biochemical components of tea and electronic tongue by support vector machine showed good prediction results. Therefore, the electronic tongue is a useful tool for Congou black tea classification.

## 1. Introduction

Black tea is the most commonly consumed tea type in the world. In China, there are three types of black tea, namely, Souchong black tea, Congou black tea, and broken black tea. Keemun black tea and Dianhong Congou black tea are varieties of Congou black tea. They produce very similar tea infusions that, commonly, are difficult to distinguish. They also present only subtle differences in tea grades, which does not help with their identification.

For a long time, tea grades and quality identification mainly relied on human sensory evaluation, which, however, may be biased by external and subjective factors. The traditional detection of tea biochemical components requires a lot of time and energy. To achieve a standardized and objective tea evaluation, it is necessary to introduce a digital assessment system.

The electronic tongue uses a sensor array to perform a qualitative or quantitative analysis of liquid samples through pattern recognition. It has been widely used to analyze food products, such as coffee [1], wine [2], milk [3,4], honey [5,6,7,8], juice [9], and others [10,11]. Since 2000, a growing number of studies have focused on the use of the electronic tongue for tea analysis. The main ones are listed in Table 1.

Previous studies showed that the electronic tongue is useful in tea classification, but few studies have applied it to distinguish similar types of tea. Some of the samples tested were purchased from supermarkets or retailers, were of assorted brands, had different origins, and were produced by diverse techniques. Most of these studies ignored the relationship between biochemical components of tea and electronic tongue response. In this work, we analyzed two similar types of Chinese Congou black tea, Dianhong Congou black tea, and Keemun black tea, using standard samples. Both types are characterized by seven grades. We showed the ability of the electronic tongue not only to distinguish tea types and grades but also to identify the biochemical components of different teas. Principal component analysis and discriminant analysis were used for type identification. Partial least-squares analysis, back-propagation neural network, discriminant analysis were used for grade identification. By support vector machine models, the tea biochemical compositions could be predicted by electronic tongue response.

## 2. Results

### 2.1. Biochemical Composition

The biochemical composition of samples is presented in Table 2 and Table 3. Twelve biochemical compositions were analyzed. As revealed in Table 2, the average content of water extract of Dianhong tea was 0.50% higher than that of Keemun tea. For Dianhong tea, the content of water extract changed according to the tea grade in this order: DS > D1 > D2 > D6 > D4 > D3 > D5. For Keemun tea, the content of different grades changed in this order: K2 > KS > KT > K1 > K4 > K5 > K3. On the whole, the higher the grade, the higher the content of water extract. For tea polyphenols, the average content in Dianhong tea was 3.36% higher than in Keemun tea. For Dianhong tea, the highest polyphenol content was found in D5, corresponding to 14.07%, followed by D1, with 14.04%. The polyphenol content of all the others Dianhong teas was below 14.00%, in this order D2 > D4 > D6 > D3 > DS. For Keemun tea, no sample presented a polyphenol content higher than 14%. The samples with the highest polyphenol content were K2 with 13.21% and K1 with 13.03%. The polyphenol content in the other Keemun samples followed the order K3 > KS > KT > K4 > K5. For amino acids, the average content in Keemun tea was 0.55% higher than in Dianhong tea. For Dianhong tea, the content of amino acids was below 4.00% in all samples, in the order D2 > D3 > D4 > D5 > D6 > D1 > DS. For Keemun tea, the amino acid content was over 4.00% for two grades, i.e., KT with 4.19% and K1 with 4.08%. For the other samples, the amino acid content followed the order KS > K2 > K3 > K4 = K5. For caffeine, its content in Keemun tea was 4.77% higher than in Dianhong tea. Overall, for Dianhong tea and Keemun tea, the content of caffeine varied depending on the tea grade according to the order D3 > D4 > D5 > D6 > DS > D2 > D1 and KT > KS > K3 > K4 > K5 > K1 > K2, respectively. However, the contents of tea polyphenols, amino acids and caffeine in these samples may not always be related to their grade.

Table 3 reports the contents of eight catechin monomers. The content of EGCG and ECG were the highest in both black tea types, followed by GC and EC. The content of four other catechins were all below 1 mg/g in all samples. For EGCG, the average content in Keemun tea was 41.47% higher than in Dianhong tea. The content of EGCG varied in Dianhong tea samples of different grade in the order D5 > D6 > D4 > D3 > D1 > D2 > DS, whereas in Keemun tea, it varied in the order K1 > K2 > KS > K3 > KT > K4 > K5. For ECG, the average content in Dianhong tea was 23.45% higher than in Keemun tea; in samples of different grade, the ECG content varied in the order D6 > D5 > D4 > D3 > D2 > D1 > DS, showing a direct relationship with the sample grade. For Keemun tea, the EGC content varied depending on the sample grade according to the order K1 > KS > KT > K2 > K3 > K4 > K5, increasing with the increase of the sample grade as well.

### 2.2. Sensory Evaluation Score

The results of tea sensory evaluation are shown in Table 4. Each black tea was classified according to its characteristics. In general, the sensory score related to specific features and the total score were strongly correlated to the tea grade. The higher was the grade, the higher the score. For Dianhong tea, the total score of each sample varied in the order DS > D1 > D2 > D3 > D5 > D4 > D6, while for Keemun tea, it varied in the order KT > KS = K1 > K2 > K3 > K4 > K5 > K6. An important aspect for human sensory evaluation was taste. For Dianhong tea, the taste score varied depending on the grade in the order DS > D1 > D2 > D5 > D3 > D6 > D4, while for Keemun tea, the taste scored varied in the order KT > KS > K1 = K2 > K3 = K4 > K5

### 2.3. Electronic Tongue Response

The electronic tongue response of the seven sensors varied from 0 to 20 s, and remained stable from 20 to 120 s. Therefore, the response value at 120 s was considered in the following analysis. The data in Figure 1 show the electronic tongue response values for one of the highest grades of each kind of tea (DS and KT). The trends of the responses were similar for the two tea types. The response value of the sensor ZZ for Keemun tea was higher than that for Dianhong tea. The response value of the sensor BA for Dianhong tea was higher than that for Keemun tea. The response values of the sensors JB, CA, and BB were very close for the two black tea varieties. The response value of the sensor ZZ was the highest for both tea varieties, reaching above 2000. Besides, it can be seen from Figure 2 that the response value of the sensor BA slightly varied for different grades of Dianhong tea, showing a higher value for higher grades of the sample. This trend was not so evident for Keemun tea. The response values of the sensor JB showed were quite distinct for every grade of the two types, showing a strong negative correlation with the sample grade. The response value of the sensor HA showed a similar relationship as that of the sensor JB. The sensor CA, on the contrary, showed a higher response value with increasing grade.

### 2.4. Principal Component Analyses (PCA)

We used PCA to analyze the electronic tongue data from the two varieties of black tea. The results are displayed in Table 5. Combined with the principle that the eigenvalue should be greater than 1, two principal components were extracted. The cumulative contribution rate was 80.48%, which can be considered to represent most of the information of the original variable. The sensors BB, GA, and CA contributed to a large extent to the first principal component, while the sensor ZZ contributed the most to the second principal component.

The scatter plot in Figure 3 shows the relationship between the variables’ distribution. Two principal components were plotted on the X and Y axes to obtain a two-dimensional (2-D) scatter plot (Figure 3a). The samples of the two type distributed into two distinct areas. Keemun tea samples located on the negative side of the PC2 axis, whereas Dianhong tea samples located on the positive side of the PC2 axis.

Meanwhile, the higher the sample position along the PC1 axis, the lower the sample grade. The grades DS and KT could be clearly distinguished from the other grades. Teas of grades D1 to D6 distributed in a distinctive manner. However, there were overlaps between samples of adjacent grades, i.e., between samples D1 and D2, D3 and D4, D4 and D5, and D5 and D6. K2 samples partially overlapped with K1 and K4 samples, K3 samples overlapped with K4 and K5 samples, which could not be distinguished from each other.

The first three principal components represented 94.02% of the information of the original variables, with the three-dimensional (3-D) diagram drawn on the basis of their score values. The trend (Figure 3b) was roughly comparable to that of the 2-D scatter plot. Both black tea varieties could be distinguished in 3-D scatter plot. Nonetheless, for each type, only DS, D1, KT, and KS samples could be recognized, and it was difficult to identify samples of other grades. Therefore, further discrimination was necessary.

### 2.5. Discriminant Analysis (DA)

Fisher discriminant analysis was used to process the responses of the seven sensors. The extraction of the first two variables represented 96.20% of the overall sample information, which was better than what obtained with the PCA. The primary variables represented 77.50% of the original data, and the secondary variables represented 18.70%. As can be seen from Figure 4, fisher discriminant analysis succeeded in distinguishing the two types of black tea. The positive region on the Function 1 axis characterized Keemun tea samples, while the negative area characterized Dianhong tea samples. Higher values on the Function 2 axis corresponded to lower grades of the samples. Furthermore, all samples were arranged according to their grades and closely clustered around the centroids. By cross validation (Table 6), the overall discrimination accuracy of the samples was 98.2%. Judgment accuracy for DS, D1, D2, D3, and D6 was 100%, whereas for D4 and D5, it was 91.7%. Also, one D4 sample was misjudged for D3, and one D5 sample was misjudged for D6. For Keemun tea, there was only one misjudgment, i.e., one K3 sample was misjudged for K4, the other grades were identified correctly.

### 2.6. Partial Least-Squares (PLS) Analysis

The partial least-squares model could further predict the grade of different kinds of black tea samples. When establishing the model, the response values of the seven electronic tongue sensors ZZ, BA, BB, CA, GA, HA, and JB were taken as the independent variables, while the grade was considered as the dependent variable. Ten response values for each grade were randomly chosen as the training set. Three principal components were extracted from Dianhong tea, and four principal components were extracted from Keemun tea. The PLS models are shown in Figure 5. The prediction coefficient of determination (R^2^) of PLS training model of Dianhong tea was 0.968. The root mean square error prediction (RMSEP) was 0.3286. As for Keemun tea, the prediction R^2^ was 0.964, and the RMSEP was 0.3462. Two samples of each grade were randomly selected to verify the model further. The testing R^2^ for Dianhong tea and Keemun tea were 0.960 and 0.954, respectively (Table 7). It showed high accuracy and stability of PLS model prediction.

### 2.7. Back-Propagation Neural Network (BPNN)

To investigate the grade prediction ability of the electronic tongue, a back-propagation neural network was applied. In total, 168 electronic tongue response values were randomly divided into a training set, prediction set, and retention set, in the proportion 6:3:1. The response signals of the seven sensors were inputted as input nodes. The neurons in the output layer were the grades of the two kinds of black tea. The hidden layer was one layer and contained eight neurons. Therefore, the network structure was 7–8–14. The classification accuracy was 100% for training and 72.00% for testing, while the final prediction accuracy for all samples was 95.00% (Table 8).

### 2.8. Support Vector Machine (SVM)

The support vector machine method was used to construct a quantitative prediction model of tea biochemical components. This method can predict the content of tea biochemical components only on the basis of the response value of the electronic tongue. We randomly selected 75% of samples as the training set and used the rest as the prediction set for support vector machine modeling. The correlation coefficient is usually considered to evaluate the performance of models. Rc represents the correlation coefficient of the correction set, and Rp represents the correlation coefficient of the prediction set. The root-mean-square error (RMSE) illustrates the degree of dispersion of the samples. RMSEC represents the dispersion degree of the correction set, and RMSEP represents the dispersion degree of the prediction set. Combining the data in Figure 6 and Table 9, the best parameters c and g of the support vector machine model for the Tea polyphenols-electronic tongue were c = 5.6569, g = 1, Rc = 0.9786, RMSEC = 0.0015, Rp = 0.9951, and RMSEP = 0.0006. Figure 6 shows that the original data of the training set and testing set samples largely overlapped with predicted data, which indicated that the model performed well. The best parameters could also be found for other biochemical components, as shown in Table 9. In this experiment, the correlation coefficients of the training set model were all above 0.9636, and those of the testing set were all above 0.7022. The RMSE was no higher than 0.0241, showing that the support vector machine model could achieve a good prediction.

## 3. Discussion

In this paper, the response values of different electronic tongue sensors for Congou black tea varied, with ZZ > JB > BB > BA. The response values of the sensors HA, GA, and CA were lower than those of the sensor BB but higher than those of the sensor BA. They varied depending on the samples’ grade (Figure 1 and Figure 2). Our results are different from those reported for green tea using the same electronic tongue [16,17,26]. Chen et al. [17] reported that the green tea response values of the different sensors varied in the order BA > CA > ZZ > JB > BB > HA > GA. Xiao et al. [26] pointed out that the electronic tongue response values for Xihulongjing (a Chinese green tea) from the different sensors were in the order CA > ZZ > GA > JB > HA > BB > BA. Rodrigues et al. [27] found that green tea presented three oxidation peaks, while black tea had two. Differences between green tea and black tea were also found using other types of electronic tongues because the processing technology of black tea and green tea is different, leading to significant differences in the biochemical content of the teas. Green tea is unleavened, it contains more polyphenols and amino acids. Black tea is fully fermented, with a low content of polyphenols, EGCG, EGC. Tea polyphenols and catechins mainly provide astringency, acerbity, and bitterness. Amino acids connected to the fresh and brisk taste, and others can also have bitter, sweet or sour taste. Caffeine provides the bitter taste. They are responsible for the fresh and brisk taste of green tea and the sweet and mellow taste of black tea [15,27,28]. Different quantities of the components lead to different tea flavors and, consequently, to different response values. From the radar map (Figure 2), the electronic tongue for both kinds of teas tended towards similar. Because the processing of two Congou black teas was the same and consisted of withering, rolling, fermenting and drying. Therefore, the response values determined may be typical of Chinese Congou black tea.

The electronic tongue response combined with the PCA and the DA could distinguish the two types of Congou black tea. The scatter plots of PCA (Figure 3) and the DA (Figure 4) showed that the samples of the two types of tea were situated in two areas, which clearly distinguished one kind from the other. The DA performed better. Although these samples were all processed according to the Congou black tea technology, they came from different tea varieties; also, there were certain small differences in their processing. For example, the strength and time of rolling changed the amount of polyphenols that can be dissolved. The degree of oxidation was also different. There was a marked difference in the biochemical composition of the two types of tea (Table 2 and Table 3
Table 2; Table 3). The contents of water extract, EC, and ECG of Dianhong tea were higher than those of Keemun tea, but the contents of caffeine, EGCG, GCG, and EGC in Kemmun tea were higher than in Dianhong tea. Furthermore, the average response values (Figure 2) of the sensor ZZ and BA varied significantly for the two types of tea. The value of the sensor ZZ for Dianhong tea was lower than that for Keemun tea, and the value of the sensor BA for Dianhong was higher than that for Keemun tea. The electronic tongue sensor ZZ is sensitive to sour and sweet tastes, while the sensor BA is sensitive to the sour taste [11]. The sweet-tasting substances in tea mainly include soluble sugar and certain amino acids. Sour substances form in black tea during fermentation, and mostly include some amino acids, organic acids, and gallic acids. Therefore, small differences in the fermentation procedure resulted in some different response values for these two Congou black teas. For the sensory evaluation, Keemun tea was sweeter and more fresh. It can be seen that the response values of the electronic tongue were consistent with the results of sensory evaluation. The sensor ZZ and BA appeared the most effective in identifying the type of black tea.

For grade discrimination, the accuracy of classification of all samples by the DA (Table 6) and the BPNN (Table 8) was greater than 95.00%. Li et al. [20] had previously successfully classified Tieguanyin (a kind of Chinese oolong tea) by BPNN with good results. Mondal et al. [29] used a pseudo-outer-product-based fuzzy neural network to classify four black tea grades. The sample distribution of the PCA (Figure 3) and the DA (Figure 4) showed sample distribution according to the tea grades. This reflected the intrinsic distinction of tea. From Table 2 and Table 3, the content of water extract, amino acids, CG, and GC were higher in higher-grade. Other biochemical content varied in different grades [30]. In terms of human sensory evaluation (Table 4), a lower taste score corresponded to lower-grade samples. Gao et al. [31] previously used PCA to identify Pu -erh tea (a kind of Chinese dark tea) and showed that each grade could be distinguished by PCA. However, in this study, the PCA scatter plot clearly distinguished only the highest grade (Figure 3). The highest- grade samples of each tea type arranged loosely in the scatter plot without overlapping with other samples. Loose distribution of PCA scatter plot were also observed in other grade classification studies [18,19,26,32,33,34]. Because extraction comparison of PCA was based on the original data rather than the category [32]. The DA region graph (Figure 4) showed that different grades could be distinguished well and samples of the same grade clustered in the centroids, without overlapping with other samples. In Figure 4, higher values of the Function 2 axis corresponded to lower grades. The PLS model was used to predict the grade of each sample of tea. The R^2^ of two tea types were both above 0.95, and the prediction results for Dianhong tea were better. In addition, it can be seen from the radar map (Figure 2) that the responses of the sensor JB, HA, and CA varied significantly for different tea grades. The response values of the sensor JB and HA decreased for samples of higher grades. The sensor CA showed the opposite trend. The sensor JB and HA are sensitive to the sour taste, while the sensor CA is sensitive to both the sour taste and the sweet taste. Generally, high-grade black tea has a pleasant sweet mellow taste, low-grade black tea presents bland taste. Combined with the sensory evaluation (Table 4), the overall score and quality of tea taste decreased with grade reduction.

## 4. Materials and Methods

### 4.1. Sample Preparation

Standard samples of Dianhong Congou black tea and Keemun black tea corresponding to seven grades were obtained from Yunnan and Anhui provinces. Dianhong Congou black tea of special grade and grade of Grades 1 to 6 were marked as DS, D1, D2, D3, D4, D5, and D6. Keemun black tea classified as into Teming, special grade, and Grade 1 to Grade 5, indicated as KT, KS, K1, K2, K3, K4, and K5. All samples were packed with brown paper and stored at 4 °C before testing.

### 4.2. Biochemical Composition Detection

The tea biochemical composition was detected according to the National Standards of the People’s Republic of China (modified ISO 1573: 1980, ISO 9768: 1994, ISO 10727: 1995, ISO 14502-1/2: 2005 and GB/T 8314-2013). Twelve items were analyzed: water extract, total tea polyphenols, amino acids, caffeine, catechin (C), epicatechin (EC), gallocatechin (GC), epigallocatechin (EGC), catechin gallate (CG), epicatechin ga1late (ECG), gallocatechin gallate (GCG), and epigallocatechin gallate (EGCG).

### 4.3. Tea Sensory Evaluation

Tea sensory evaluation was performed following the National Standards of the People’s Republic of China (GB/T 23776-2018) by professional tea tasters. The grading standard for congou black tea was based on a total score of 100, with 25% accounting for the appearance of dry tea, 10% for liquor color, 25% for aroma, 30% for taste, and 10% for infused leaves.

### 4.4. Electronic Tongue and Data Acquisition

A potentiometry electrochemical electronic tongue (α-Astree II; Alpha MOS company, Toulouse, France) was employed in this experiment. It included an array of seven chemical sensors (ZA, BB, BA, GA, HA, JB, and CA) and one Ag/AgCl reference electrode. Each sensor was coated with a different molecular film, which had different adsorption properties, discriminating different tastes. Each sensor had cross-sensitivity to bitterness, savory, saltiness, sourness, and sweetness. The potentiometric difference between the sensor and the Ag/AgCl reference electrode was recorded as the response signal. Tea preparation for electronic tongue detection was as follows: 3 g of dry tea was brewed in 150 mL of freshly boiled pure water for 5 min, and the tea infusion was then carefully separated into an empty bowl. The filtered infusion was allowed to cool down for at least 30 min until it reached room temperature (25 °C). After then, 80 mL of infusion was transferred into a special beaker for electronic tongue detection. In this experiment, each sample was tested for 120 s in order to obtain consistent results. The software recorded the response value of each sensor per second, and the last second of data was used as the original data for statistical analysis. Each sensor was cleaned for 15 s during the continuous test of the same sample, and for 120 s before testing another sample in order to prevent cross-contamination between samples and ensure the cleanliness of the sensor [26].

### 4.5. Data Analysis

One-way analysis of variance was used to test the variance of tea biochemical components in the different samples of the two tea types. Principal component analysis derived a few principal components from the original variables so that they retained as much information as possible about the original variables. Discriminant analysis and back-propagation neural network were used to reflect the accuracy of the type and grade of electronic tongue prediction. Partial least-squares analysis showed the grade prediction effect for each type. The support vector machine model further confirmed the relationship between electronic tongue response values and tea biochemical components. The figure reposting the electronic tongue response values and the radar map were obtained using Origin 2017; one-way analysis of variance, principal component analysis, discriminant analysis and back-propagation neural network were performed by SPSS 25.0, while partial least-squares analysis and support vector machine were performed using Matlab 2019a.

## 5. Conclusions

In this paper, we examined the electronic tongue response for Congou black tea. The response values of the electronic tongue’s sensors for Congou black tea varied according to the order ZZ > JB > BB > HA/GA/CA > BA (the response values of the sensors HA, GA, CA varied among samples). The response values of the sensors ZZ and BA were related to the type of Congou black tea. The sensor ZZ showed a higher response value for Keemun tea, and the values recorded for Keemun tea samples at different grades were all above 2200. The sensor BA showed lower response values for Keemun tea, which were all below 500. The response values of the sensor BA for Dianhong tea were all above 500. The electronic tongue combined with PCA and DA could successfully distinguish the two tea types. For grade discrimination, the response values of the sensors JB and HA decreased for high-grade samples, while the response value of the sensor CA increased as the sample grade increased. The electronic tongue combined with PLS, DA, and BPNN could effectively identify the tea grade. Among the different pattern recognitions obtained, the DA region graph could effectively discriminate both type and grade simultaneously. For quantitative analysis, the SVM model showed a good correlation between the seven electronic tongue sensors’ response values and the biochemical components of Congou black tea. Therefore, this model can predict the content of the biochemical components of tea on the basis of the response values of the electronic tongue. From the above analysis, the electronic tongue could serve as an effective tool for classification of similar Congou black teas.

## Figures and Tables

**Figure 1 molecules-24-04549-f001:**
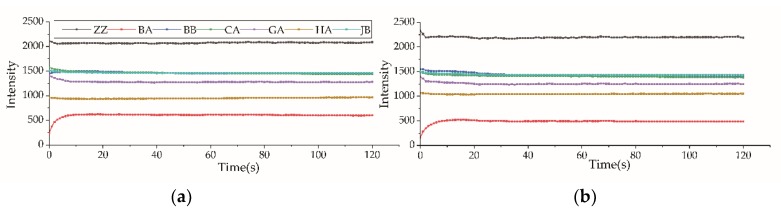
Electronic tongue response for black tea. (**a**) Electronic tongue response value for Dianhong tea, DS grade; (**b**) Electronic tongue response value for Keemun tea, KT grade. ZZ, BA, BB, CA, GA, HA, and JB: electronic tongue sensors.

**Figure 2 molecules-24-04549-f002:**
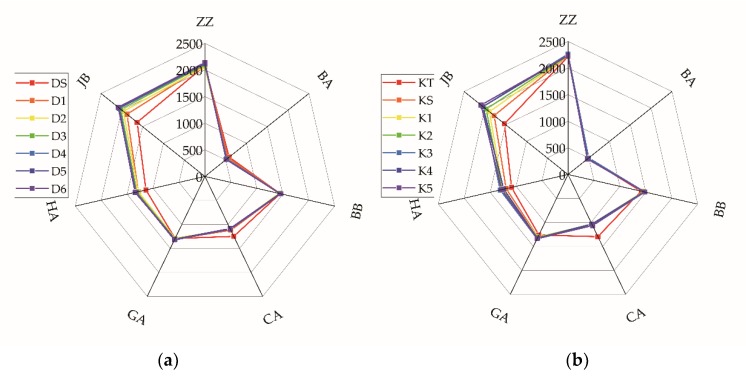
Average response value of the electronic tongue for black tea. (**a**) Dianhong tea; (**b**) Keemun tea.

**Figure 3 molecules-24-04549-f003:**
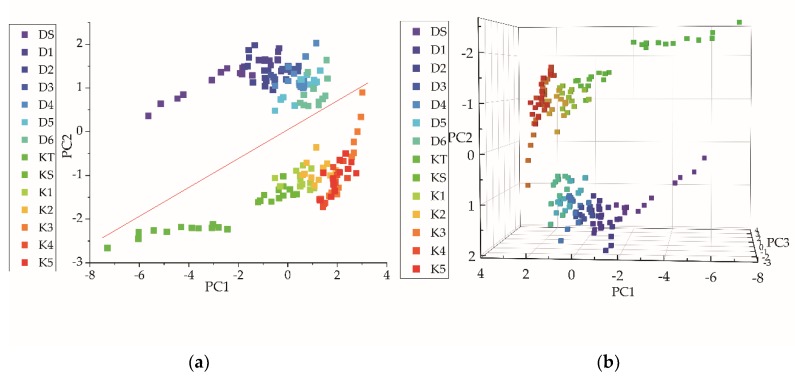
Scatter plots of principal component analysis (PCA) of electronic tongue data from the two types of black tea. (**a**) 2-D scatter plot; (**b**) 3-D scatter plot.

**Figure 4 molecules-24-04549-f004:**
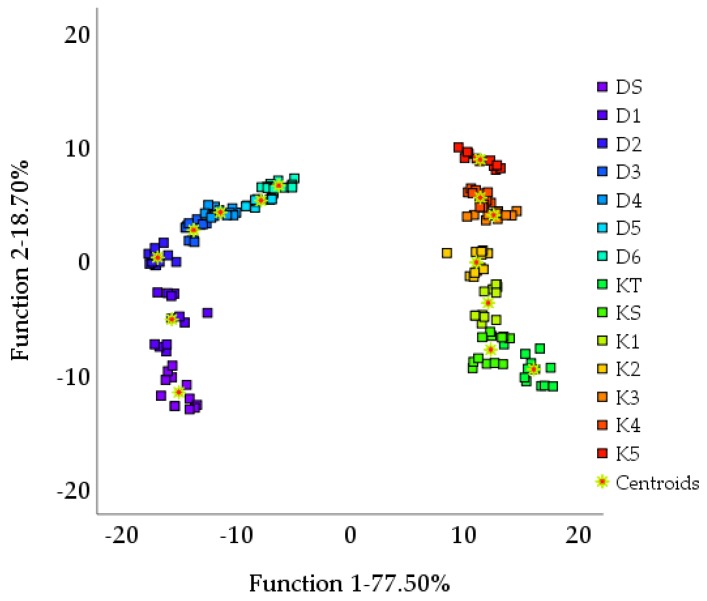
Discriminant analysis (DA) region graph of black tea samples.

**Figure 5 molecules-24-04549-f005:**
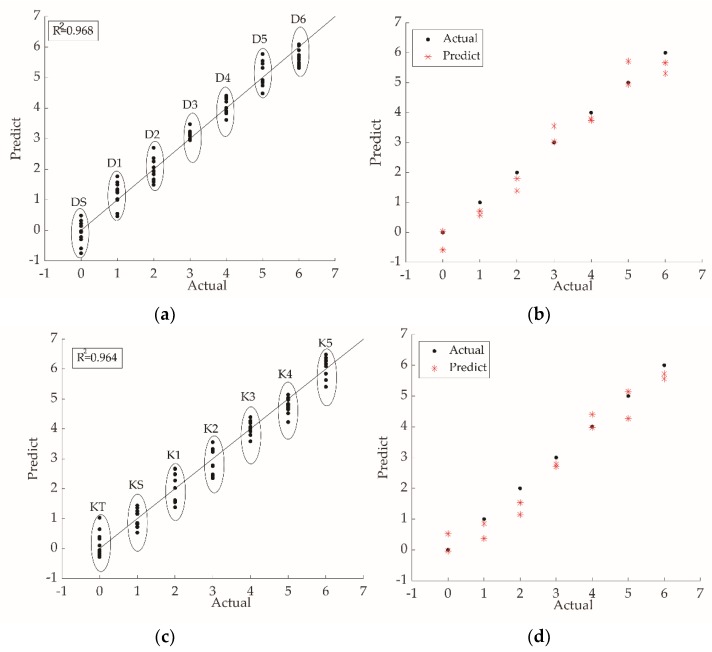
Partial least-squares (PLS) model diagram of two types of black tea. (**a**) Training set for Dianhong tea; (**b**) testing set for Dianhong tea; (**c**) training set for Keemun tea; (**d**) testing set for Keemun tea.

**Figure 6 molecules-24-04549-f006:**
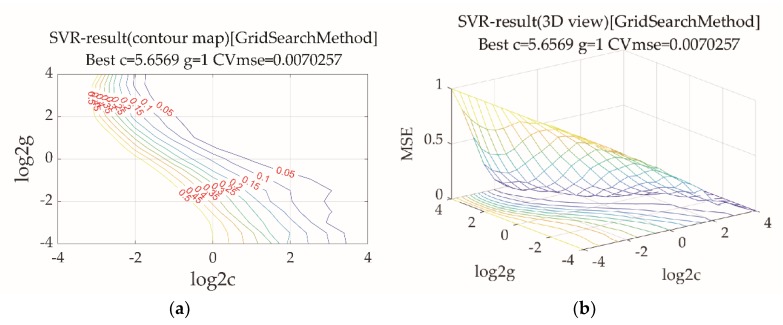
Electronic tongue quantitative prediction model parameters for tea polyphenols. (**a**) SVR result counter map; (**b**) SVR result 3D view; (**c**) training set regression predict by SVM; and (**d**) testing set regression predict by SVM; SVR: Support vector regression; SVM: support vector machine.

**Table 1 molecules-24-04549-t001:** Main studies results of the use of the electronic tongue for tea analysis since 2000.

Tea Sample	Type of Electronic Tongue	Research Goal	Pattern Recognition Method	Reference
Korean green tea, British black tea	Potentiometry	Tea type identification	PCA/PCR/PLS	[12]
Indian black tea	Impedance Spectroscopy		PCA	[13]
Swedish black tea and green tea	Voltammetry		MVDA/PCA	[14]
Chinese and Vietnamese black, green, red, and white tea	Voltammetry		SOM/PCA/HCA	[15]
Chinese green tea and black tea	Potentiometry	Tea grade identification	MVDA/PCA	[16]
Chinese green tea	Potentiometry		KNN/ANN	[17]
Chinese green tea	Potentiometry		PCA	[18]
Indian black tea	Voltammetry		PCA/KNN	[19]
Chinese green tea and black tea	Potentiometry	Tea origin identification	MVDA/PCA	[16]
Chinese Tieguanyin tea	Potentiometry		ANN/PCA/HCA	[20]
Indian black tea	Voltammetry	Tea quality analysis	PCA/LDA/BP-MLP	[21]
Spanish andRussian black tea	Potentiometry		PLS	[22]
Chinese green tea	Potentiometry	Tea fraud identification	PCA/HCA/ANN	[23]
Black tea from Kenya, India, Indonesia, China, Sri Lanka, Vietnam, and etc.	Voltammetry	Tea biochemical content analysis	Si -CARS-PLS	[24]
Indian black tea	Voltammetry		ANN//VVRKFA/SVR	[25]

PCA: principal component analysis; PCR: principal component regression; PLS: partial least-squares; MVDA: multivariate data analysis; SOM: self-organizing maps; HCA: hierarchical cluster analysis; KNN: K-nearest neighbors; ANN: artificial neural network; LDA: linear discrimination analysis; BP-MLP: back-propagation multilayer perceptron; Si-CARS-PLS: synergy interval partial least square combined with competitive adaptive reweighted sampling; VVRKFA: vector-valued regularized kernel function approximation; SVR: support vector regression.

**Table 2 molecules-24-04549-t002:** Content of biochemical components in Dianhong (D) and Keemun (K) black teas.

Grade	Water Extract (%)	Tea Polyphenols (%)	Amino Acids (%)	Caffeine (%)
DS	41.12 ± 1.26 ^a^	11.54 ± 0.35 ^de^	3.49 ± 0.06 ^fg^	2.72 ± 0.14 ^cde^
D1	40.91 ± 1.01 ^a^	14.04 ± 1.19 ^a^	3.54 ± 0.05 ^fg^	2.14 ± 0.13 ^ef^
D2	40.60 ± 1.97 ^ab^	13.70 ± 0.53 ^ab^	3.93 ± 0.06 ^abcd^	2.28 ± 0.48 ^def^
D3	38.53 ± 0.59 ^de^	13.08 ± 0.43 ^abc^	3.90 ± 0.20 ^abcde^	3.65 ± 0.05 ^ab^
D4	38.90 ± 0.72 ^cde^	13.41 ± 0.39 ^abc^	3.74 ± 0.15 ^cdef^	3.55 ± 0.04 ^ab^
D5	38.13 ± 0.23 ^de^	14.07 ± 0.96 ^a^	3.68 ± 0.13 ^cdef^	3.18 ± 0.13 ^bc^
D6	39.19 ± 1.06 ^bcd^	13.09 ± 1.14 ^abc^	3.65 ± 0.19 ^def^	3.15 ± 0.01 ^bc^
KT	39.85 ± 0.78 ^abcd^	12.40 ± 0.42 ^cd^	4.19 ± 0.26 ^a^	4.17 ± 0.01 ^a^
KS	40.23 ± 0.76 ^abc^	12.42 ± 0.68 ^bcd^	3.98 ± 0.17 ^abc^	3.95 ± 0.29 ^a^
K1	39.20 ± 0.68 ^bcd^	13.03 ± 0.47 ^abc^	4.08 ± 0.12 ^ab^	2.39 ± 1.10 ^def^
K2	40.99 ± 0.53 ^a^	13.21 ± 0.35 ^abc^	3.86 ± 0.19 ^bcde^	2.05 ± 0.12 ^f^
K3	37.42 ± 0.20 ^e^	12.62 ± 0.89 ^bcd^	3.59 ± 0.21 ^ef^	3.80 ± 0.09 ^ab^
K4	38.51 ± 0.47 ^de^	12.27 ± 0.32 ^cd^	3.27 ± 0.25 ^g^	3.60 ± 0.13 ^ab^
K5	38.22 ± 0.74 ^de^	10.51 ± 0.08 ^e^	3.27 ± 0.12 ^g^	2.81 ± 0.26 ^cd^

Values are presented as the means ± standard deviation (*n* = 3). Different lowercase letters indicate significant difference.

**Table 3 molecules-24-04549-t003:** Content of catechin monomers in the two types black tea (mg/g).

Grade	GC	EGC	C	EC	EGCG	GCG	ECG	CG
DS	1.71 ± 0.10 ^d^	0.19 ± 0.00 ^f^	0.10 ± 0.01 ^fg^	0.77 ± 0.02 ^g^	3.27 ± 0.09 ^k^	0.11 ± 0.01 ^f^	3.67 ± 0.05 ^gh^	0.69 ± 0.04 ^a^
D1	1.61 ± 0.02 ^de^	0.18 ± 0.01 ^f^	0.16 ± 0.02 ^ef^	0.96 ± 0.00 ^e^	3.69 ± 0.15 ^j^	0.12 ± 0.00 ^f^	4.42 ± 0.24 ^f^	0.55 ± 0.40 ^ab^
D2	0.29 ± 0.04 ^i^	0.15 ± 0.01 ^f^	0.20 ± 0.00 ^e^	0.99 ± 0.01 ^e^	3.52 ± 0.04 ^jk^	0.10 ± 0.01 ^f^	4.48 ± 0.10 ^f^	0.07 ± 0.01 ^f^
D3	1.25 ± 0.01 ^g^	0.38 ± 0.01 ^e^	0.37 ± 0.00 ^d^	1.81 ± 0.03 ^d^	4.46 ± 0.08 ^i^	0.15 ± 0.01 ^f^	6.27 ± 0.29 ^d^	0.12 ± 0.01 ^ef^
D4	1.42 ± 0.02 ^f^	0.47 ± 0.02 ^de^	1.10 ± 0.00 ^c^	2.53 ± 0.02 ^c^	4.96 ± 0.04 ^h^	0.16 ± 0.00 ^f^	8.28 ± 0.24 ^c^	0.10 ± 0.01 ^ef^
D5	1.29 ± 0.01 ^g^	0.63 ± 0.02 ^bc^	1.51 ± 0.03 ^b^	3.65 ± 0.09 ^b^	5.80 ± 0.09 ^g^	0.17 ± 0.00 ^f^	9.86 ± 0.05 ^b^	0.10 ± 0.00 ^ef^
D6	1.22 ± 0.02 ^g^	0.70 ± 0.00 ^abc^	1.62 ± 0.01 ^a^	4.06 ± 0.15 ^a^	5.69 ± 0.01 ^g^	0.18 ± 0.02 ^f^	11.37 ± 0.22 ^a^	0.11 ± 0.01 ^ef^
KT	2.86 ± 0.01 ^a^	0.72 ± 0.01 ^ab^	0.14 ± 0.02 ^ef^	0.61 ± 0.01 ^h^	8.15 ± 0.28 ^de^	0.53 ± 0.05 ^a^	4.61 ± 0.26 ^f^	0.40 ± 0.00 ^bcd^
KS	2.29 ± 0.07 ^b^	0.70 ± 0.06 ^abc^	0.15 ± 0.01 ^ef^	0.77 ± 0.04 ^g^	8.92 ± 0.33 ^c^	0.49 ± 0.06 ^ab^	4.66 ± 0.41 ^f^	0.41 ± 0.07 ^bcd^
K1	2.27 ± 0.05 ^b^	0.79 ± 0.02 ^a^	0.16 ± 0.01 ^ef^	1.00 ± 0.07 ^e^	9.96 ± 0.22 ^a^	0.49 ± 0.00 ^ab^	5.11 ± 0.15 ^e^	0.42 ± 0.01 ^bcd^
K2	1.99 ± 0.02 ^c^	0.69 ± 0.01 ^abc^	0.11 ± 0.01 ^f^	0.90 ± 0.03 ^ef^	9.35 ± 0.07 ^b^	0.43 ± 0.02 ^bc^	4.57 ± 0.10 ^f^	0.45 ± 0.01 ^bc^
K3	1.67 ± 0.06 ^d^	0.63 ± 0.12 ^bc^	0.04 ± 0.00 ^g^	0.82 ± 0.06 ^fg^	8.27 ± 0.12 ^d^	0.37 ± 0.06 ^cd^	3.75 ± 0.06 ^g^	0.26 ± 0.03 ^cdef^
K4	1.51 ± 0.23 ^ef^	0.72 ± 0.26 ^ab^	0.09 ± 0.10 ^fg^	0.79 ± 0.13 ^fg^	7.90 ± 0.34 ^e^	0.33 ± 0.11 ^de^	3.33 ± 0.15 ^h^	0.32 ± 0.16 ^cde^
K5	0.96 ± 0.04 ^h^	0.55 ± 0.02 ^cd^	0.12 ± 0.09 ^f^	0.73 ± 0.04 ^g^	6.72 ± 0.35 ^f^	0.27 ± 0.06 ^e^	2.62 ± 0.24 ^i^	0.21 ± 0.03 ^def^

Values are presented as the means ± standard deviation (*n* = 3). Different lowercase letters indicate significant difference.

**Table 4 molecules-24-04549-t004:** Sensory evaluation of the two types black tea.

Tea Sample	Appearance of Dry Tea	Liquid Color	Aroma	Taste	Infused Leave	Total Score
DS	95.00	95.00	92.00	93.00	93.00	93.45
D1	94.00	94.00	89.00	89.00	92.00	91.05
D2	92.00	93.00	90.00	88.00	92.00	90.40
D3	90.00	93.00	90.00	85.00	90.00	88.80
D4	88.00	89.00	86.00	80.00	88.00	85.20
D5	87.00	89.00	90.00	86.00	88.00	87.75
D6	85.00	88.00	88.00	81.00	85.00	84.85
KT	98.00	95.00	96.00	85.00	93.00	92.80
KS	98.00	94.00	96.00	85.00	92.00	92.60
K1	97.00	93.00	95.00	87.00	92.00	92.60
K2	96.00	93.00	95.00	87.00	90.00	92.15
K3	95.00	89.00	94.00	85.00	88.00	90.45
K4	92.00	89.00	92.00	80.00	88.00	87.70
K5	90.00	88.00	92.00	80.00	85.00	86.80

**Table 5 molecules-24-04549-t005:** Characteristic values and contributions of principal components of black tea.

Principal Components	1	2	3	4	5	6	7
Eigenvalue	3.85	1.79	0.95	0.25	0.13	0.02	0.01
Contribution rate (%)	54.96	25.52	13.54	3.63	1.87	0.30	0.18
Cumulative contribution rate (%)	54.96	80.48	94.02	97.65	99.52	99.82	100.00

**Table 6 molecules-24-04549-t006:** Discriminant analysis results for the two types of black tea.

Tea Sample	Original Data			Cross Validation		
	Number of Samples	Number of Mistakes	Accuracy%	Number of Samples	Number of Mistakes	Accuracy%
DS	12	0	100.00	12	0	100.00
D1	12	0	100.00	12	0	100.00
D2	12	0	100.00	12	0	100.00
D3	12	0	100.00	12	0	100.00
D4	12	0	100.00	12	1	91.70
D5	12	0	100.00	12	1	91.70
D6	12	0	100.00	12	0	100.00
KT	12	0	100.00	12	0	100.00
KS	12	0	100.00	12	0	100.00
K1	12	0	100.00	12	0	100.00
K2	12	0	100.00	12	0	100.00
K3	12	1	91.70	12	1	91.70
K4	12	0	100.00	12	0	100.00
K5	12	0	100.00	12	0	100.00

**Table 7 molecules-24-04549-t007:** Comparison of predictive performance between the two types black tea.

Tea samples	Training		Testing	
	RMSEP	R^2^	RMSEP	R^2^
Dianhong	0.329	0.968	0.427	0.960
Keemun	0.346	0.964	0.445	0.954

R^2^: coefficient of determination; RMSEP: root-mean-square error prediction.

**Table 8 molecules-24-04549-t008:** Back-propagation neural network results for the two types of black tea.

Tea Sample	Training			Holding		
	Number of Samples	Number of Mistakes	Accuracy (%)	Number of Samples	Number of Mistakes	Accuracy (%)
DS	7	0	100.00	2	0	100.00
D1	5	0	100.00	1	0	100.00
D2	9	0	100.00	1	0	100.00
D3	4	0	100.00	1	1	0.00
D4	3	0	100.00	3	0	100.00
D5	6	0	100.00	3	0	100.00
D6	10	0	100.00	1	0	100.00
KT	5	0	100.00	2	0	100.00
KS	8	0	100.00	1	0	100.00
K1	8	0	100.00	2	0	100.00
K2	8	0	100.00	0	0	0.00
K3	8	0	100.00	1	0	100.00
K4	10	0	100.00	0	0	0.00
K5	7	0	100.00	2	0	100.00
Total	98	0	100.00	20	1	95.00

**Table 9 molecules-24-04549-t009:** Support vector machine model parameters for biochemical components of black tea.

Model	Training		Testing	
	Rc	RMSEC	Rp	RMSEP
Water extract-electronic tongue	0.9916	0.0008	0.7022	0.0241
Tea polyphenols-electronic tongue	0.9786	0.0015	0.9951	0.0006
Amino acids-electronic tongue	0.9816	0.0019	0.9546	0.0083
Caffeine-electronic tongue	0.9887	0.0013	0.8886	0.0159
GC-electronic tongue	0.9636	0.0027	0.8969	0.0085
EGC-electronic tongue	0.9721	0.0023	0.9856	0.0020
C-electronic tongue	0.9850	0.0018	0.9910	0.0005
EC-electronic tongue	0.9944	0.0006	0.9883	0.0006
EGCG-electronic tongue	0.9891	0.0011	0.9923	0.0011
GCG-electronic tongue	0.9830	0.0019	0.9884	0.0025
ECG-electronic tongue	0.9946	0.0005	0.9874	0.0007
CG-electronic tongue	0.9764	0.0034	0.9597	0.0044

Rc: correlation coefficient of the correction set; Rp: correlation coefficient of the prediction set; RMSEC: root-mean-square error of the correction set; RMSEP: root-mean-square error of the prediction set.

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
