# Peer review of "Identification of Similar Chinese Congou Black Teas Using an Electronic Tongue Combined with Pattern Recognition"

_molecules, 2019, doi:10.3390/molecules24244549_

Round 1
Reviewer 1 Report
The manuscript molecules-650547 entitled “Identification of Similar Chinese Congou Black Tea Using Electronic Tongue Combined with Pattern Recognition” presents an interesting research work on an agri-food application. However, the array of sensors used is not described at any time. This is a very important weakness of the method, since it is not known if the sensors are commercial or developed by the authors. It is not explained in the main text if they are electrochemical, optical, mass or other type of sensors. The nomenclature used by the authors (sensors ZZ, HA, JB, CA, BB, GA, BA) is incomprehensible and disconcerting. This point must be corrected before accepting the article.
In addition, Tables 1 and 2 must indicate in their captions what the ± of the values means. Is it a standard deviation or a confidence interval at 95%? How many replicates?.
Having said that, the article has some editing flaws:
The Figures 5 a) and b) are missing. The lines 271-273 must be removed. Please, increase the font size of the figures (axes, legends, numbers) in general, because it is practically impossible to read.Author Response
请参阅附件。

Reviewer 2 Report
Presented research shows identifiaction of black tea types and grades using electronic tongue combined with different chemometric methods (i.e. PCA, DA, BPNN, PLS and SVM. As Authors wrote: the study not only showed the ability of electronci tongue for distinguihing types of tea bau also the relationship between e-tongue response and biochemical composition of tea (and other parameters determined using standard methods).
Reviewer asks authors to response the following comments:
The authors used in the research an electronic toungue, but in the text (in par. Materials and Methods) there is no detailed information about types of the sesnors used and construction of these device. Please also add information abour signal processing in the e-tongue system. How the data for statistical analysis are prepared?In Introduction section: lines 36-46 should be placed in form of the table. They will be presented more clearly. In section 2.3. Authors use abbreviations linked with sensors (i.e. JB, CA, BB). What does the abbreviations mean? In Figure 1 the y-axis is described as "intensity". What is the meaning of this intensity? In line 97 Authors wrote: "ZZ and HA showed higher response values 97 for Keemun tea than for Dianhong tea, while sensor BA showed just the opposite". This statement does not follow from the Figure 2. Please explain. In Figure 3 please use the same symbols for samples at plot (a) and (b). The same goes for the others Figures. The parameter R2 is incorrectly defined as the correlation coefficient. Please define abbreviations: RMSEP, RMSEC, RMSE, Rc, Rp. They must be defined/explained at least once in the text Figure 6 is completely legible. Please select only the most important charts/plots and discuss them, not copy everything from the program
7 lines of Conclusions for such a large area of research it is definitely not enough. Please extend this section.
In my opinion, presented article can be reconsider after major revision.
Round 2
Reviewer 2 Report
All the reviewer's suggestions were included in the text. Currently the manuscript can be accepted in its present form for further stages of evaluation.